# Organic Selenium (OH-MetSe) Effect on Whole Body Fatty Acids and *Mx* Gene Expression against Viral Infection in Gilthead Seabream (*Sparus aurata*) Juveniles

**DOI:** 10.3390/ani11102877

**Published:** 2021-09-30

**Authors:** Yiyen Tseng, David Dominguez, Jimena Bravo, Felix Acosta, Lidia Robaina, Pierre-André Geraert, Sadasivam Kaushik, Marisol Izquierdo

**Affiliations:** 1Aquaculture Research Group (GIA), Institute of Sustainable Aquaculture and Marine Ecosystems (ECOAQUA), Universidad de Las Palmas de Gran Canaria, Crta. Taliarte s/n, 35214 Telde, Spain; david.dominguez103@alu.ulpgc.es (D.D.); jimena.bravo@ulpgc.es (J.B.); felix.acosta@ulpgc.es (F.A.); lidia.robaina@ulpgc.es (L.R.); sachi.kaushik@ulpgc.es (S.K.); marisol.izquierdo@ulpgc.es (M.I.); 2Adisseo France S.A.S., 10 Place du General de Gaulle, Antony, 92160 Paris, France; pierre-andre.geraert@adisseo.com

**Keywords:** gilthead seabream, mineral nutrition, organic selenium, hydroxy-selenomethionine

## Abstract

**Simple Summary:**

Dietary hydroxy-selenomethionine (OH-SeMet) reduces oxidative stress and modulates immune response against bacterial infection in fish. However, the effect of OH-SeMet on essential fatty acids with a high oxidation risk or on the response against viral infection has not been sufficiently studied. This study aimed to assess the impact of dietary OH-SeMet supplementation on whole-body fatty acid profiles and response against viral infection. Gilthead seabream (*Sparus aurata*) juveniles were fed for 91 days with three experimental diets, a control diet without Se supplementation (0.29 mg Se kg diet^−1^) and two diets supplemented with OH-SeMet (0.52 and 0.79 mg Se kg diet^−1^). Afterwards, a crowding stress challenge and an anti-viral response challenge were conducted. Selenium (Se), proximate and fatty acid composition of diets and body tissues were analyzed, as well as plasma cortisol and the antiviral response protein Mx gene expression. Elevation in dietary Se (from 0.29 to 0.79 mg kg^−1^) proportionally raised Se contents in body tissues (from 0.79 to 1.35 mg kg^−1^), increased lipid contents in whole body (from 9.46 to 10.83%), and promoted the retention and synthesis of n-3 polyunsaturated fatty acids (from 44.59 to 72.91%), reducing monounsaturated (from 44.07 to 42.00 %) and saturated fatty acids (29.77 to 26.92 %) contents in whole-body lipids. Additionally, it increased 2 h post-stress plasma cortisol levels and after poly I:C injection up-regulated *Mx* and other immune response related genes, showing, for the first time in gilthead seabream, the importance of dietary Se levels on antiviral defense.

**Abstract:**

The supplementation of fish diets with OH-SeMet reduces oxidative stress and modulates immune response against bacterial infection. However, despite the importance of essential polyunsaturated fatty acids in fish nutrition and their high risk of oxidation, the potential protective effect of OH-SeMet on these essential fatty acids has not been studied in detail. Moreover, while viral infection is very relevant in seabream production, no studies have focused the Se effects against viral infection. The aim of the present study was to assess the impact of dietary supplementation with OH-SeMet on gilthead seabream fatty acid profiles, growth performance and response against viral infection. Gilthead seabream juveniles (21.73 ± 0.27 g) were fed for 91 days with three experimental diets, a control diet without supplementation of Se (0.29 mg Se kg diet^−1^) and two diets supplemented with OH-SeMet (0.52 and 0.79 mg Se kg diet^−1^). A crowding stress test was performed at week 7 and an anti-viral response challenge were conducted at the end of the feeding trial. Selenium, proximate and fatty acid composition of diets and body tissues were analyzed. Although fish growth was not affected, elevation in dietary Se proportionally raised Se content in body tissues, increased lipid content in the whole body and promoted retention and synthesis of n-3 polyunsaturated fatty acids. Specifically, a net production of DHA was observed in those fish fed diets with a higher Se content. Additionally, both monounsaturated and saturated fatty acids were significantly reduced by the increase in dietary Se. Despite the elevation of dietary Se to 0.79 mg kg^−1^ not affecting basal cortisol levels, 2 h post-stress plasma cortisol levels were markedly increased. Finally, at 24 h post-stimulation, dietary OH-SeMet supplementation significantly increased the expression of the antiviral response myxovirus protein gene, showing, for the first time in gilthead seabream, the importance of dietary Se levels on antiviral defense.

## 1. Introduction

Selenium (Se) is an essential element in all living organisms, as it plays a crucial role for growth, antioxidant protection, and preventing or decreasing the harmful effects of stress [1]. During the aerobic cellular respiration or defense mechanisms, reactive oxygen species (ROS) are continuously generated as a byproduct. However, the excess level of unstable ROS may cause oxidative damage to proteins, nucleic acids and lipids, which can even lead to cell death processes [2]. Among several effective antioxidant enzymes, glutathione peroxidases (gpx) are a group of Se-dependent enzymes that protect cells against oxidative stress. Hence, adequate supplementation of Se has an important role in the antioxidant defense system to avoid oxidative damage in fish. Dietary Se deficiency causes growth reduction, high mortality, low gpx activity, increased risk of lipid peroxidation or altered immune-related parameters in juveniles of different species of fish: grouper (*Epinephelus malabaricus*) [3], tilapia (*Oreochromis niloticus*) [4], cobia, (*Rachycentron canadum*) [5], yellowtail kingfish (*Seriola lalandi*) [6] or gilthead seabream [7]. Similar Se deficiency symptoms are also found in marine fish larvae, such as European sea bass (*Dicentrarchus labrax*) [8] or gilthead seabream [9]. On the other hand, excessive dietary Se may cause toxicity, as Se associates with sulfur-containing amino acids cysteine (Cys) and methionine (Met), where it is translationally incorporated as selenocysteine (SeCys) and selenomethionine (SeMet) in the protein. Excessive production of these compounds may cause misincorporation into proteins, leading to potentially toxic products that may alter the Se-dependent enzymes [10], causing alterations in the defense system against oxidative stress.

Among other nutrients, polyunsaturated fatty acids (PUFA) are very sensitive to oxidation, leading to the production of compounds such as fatty acid hydroperoxides, fatty acid hydroxides, aldehydes and hydrocarbons that may be toxic, damaging membrane lipids, proteins and DNA [11]. Certain PUFAs, such as arachidonic acid (ARA), eicosapentaenoic acid (EPA) and docosahexaenoic acid (DHA), are essential nutrients that play very important roles in fish physiology, being structural components in the phospholipids of cellular membranes, precursors of bioactive molecules and a source of metabolic energy [11,12]. These fatty acids are also important for resistance to different types of acute stressors, such as crowding, handling, temperature or salinity [13,14], and modulate the immune system in fish [15].

Depending on the molecular form of Se supplement in the diet, the absorption and metabolic pathways in animals may change [16]. Indeed, dietary supplementation with organic Se forms, such as OH-SeMet or Se-yeast, promoted various beneficial effects in different fish species including juvenile and larval stages. For instance, it enhances the immune system by preventing the oxidative stress, through regulation of gpx, superoxide dismutase and catalase in juveniles of meagre (*Argyrosomus regius*) [17] or rainbow trout (*Oncorhynchus mykiss*) [18], as well as in larval stages of gilthead seabream [9] or European sea bass [8]. Additionally, organic Se promotes bone osteogenesis and improves the response to acute and chronic stress [19,20]. Moreover, feeding organic forms increases Se bioavailability and improves immune activity reinforcing bactericidal activity in yellowtail kingfish [6]. Several studies in fish indicated that dietary organic Se supplementation leads to reduced oxidative stress during crowding stress [18,20,21] and improved the antioxidant defense system against bacterial infection [22,23].

However, despite one of the most relevant diseases affecting gilthead seabream production is viral nervous necrosis (VNN), caused by infection of several viruses included in the genus *Betanodavirus*, no studies have been focused on the Se effects against viral infection in gilthead seabream. A primary defense against viral infections in fish includes the production of cytokines, small proteins important in cell signaling produced as immune responses to different pathogens. There are three functional categories of cytokines: those that regulate the innate response, those that regulate the adaptive response, and those that stimulate hematopoiesis [24]. Among the different cytokines, interferon (IFN) plays a major role in defense against viral infections in vertebrates, inhibiting viral replication and favoring apoptosis of virus-infected cells. Two types of IFN are recognized: IFN type I and type II [25]. IFN type I (IFN-I), homologous to IFNα/β in mammals, is induced by viruses in most cells. IFN-I is a key component against viral infections, inducing hundreds of genes, some of which encode direct antiviral effectors such as the GTP-binding Mx protein. Mx proteins belong to the superfamily of GTPases involved in intracellular membrane remodeling and intracellular trafficking. They interfere with viral replication in different states by inhibiting viral polymerase in the nucleus and by binding viral components in the cytoplasm [26].

In a previous study, supplementing diets for gilthead seabream juveniles with OH-SeMet at levels of 0.2 mg kg^−1^ (1.1 mg Se kg^−1^) and 0.5 mg kg^−1^ (1.4 mg Se kg^−1^) effectively increased Se contents in liver and muscle, reducing oxidative stress [20]. Despite polyunsaturated fatty acids being very prone to oxidation, such a study did not inspect the potential protective effect of OH-SeMet on these essential fatty acids for marine fish. Moreover, that previous study did not aim to demonstrate the potential effect on anti-viral defense responses, such as Mx proteins. Additionally, the Se levels tested were above the maximum Se contents in animal feeds recommended by the European Food Safety Authority (EFSA) (0.5 mg kg^−1^ [27]) and the maximum recommended supplementation of selenomethionine (0.2 mg kg^−1^ [28]). Therefore, the aim of the present study was to assess the impact of dietary supplementation with moderate levels of OH-SeMet following EFSA recommendations on fatty acid composition and to determine their potential effect on defense response against viral infections in juvenile gilthead seabream.

## 2. Materials and Methods

All the animal experiments were performed according to the European Union Directive (2010/63/EU) and Spanish legislation (Royal Decree 53/2013) on the protection of animal for scientific purposes at ECOAQUA Institute of University of Las Palmas de Gran Canaria (Canary Island, Spain).

### 2.1. Fish and Feeding Trial

Gilthead seabream juveniles were obtained from a natural spawning from the broodstock at the facilities of the Aquaculture Research Group (GIA, Gran Canaria, Spain). Fish were randomly distributed in 9 cylindroconical fiberglass tanks with a 200 L capacity, at a density of 37 fish tank^−1^ supplied with continuously running seawater and constant aeration. Triplicate groups of fish (initial weight: 21.73 ± 0.27 g) were manually fed with approximately 3% of body weight in four times per day (9:00 a.m., 12:00 p.m., 2:00 p.m. and 4:00 p.m., 6 days per week) for 91 days (13 weeks). The uneaten feed was immediately collected by siphon, then dried and weighed to determine the feed intake during the feeding period. The photoperiod was kept at 12 h light: 12 h dark. The temperature and dissolved O_2_ were measured daily with a multiparametric probe (Oxy Guard, Zeigler Bros, Gardners, USA), whereas pH was weekly measured with a pH meter (GLP 21+, CRISON, Barcelona, Spain). Water temperature, dissolved oxygen and pH along the whole trial were 22 ± 1 °C, 5.4 ± 0.0 mg/L and 7.8 ± 0.1, respectively. Seawater from the tanks was sampled and collected in small tubes to determine Se concentration at four different points during the feeding trial. Water samples were taken from all the tanks in two occasions: one at the beginning of the feeding trial before starting to feed the fishes and a second one 4 days after the beginning of the trial, 30 min after feeding. Water samples were stored in a cool, dark room until analysis.

### 2.2. Experimental Diets

The formulation, proximate, and fatty acid composition of the experimental diets are shown in Table 1 and Table 2, respectively. Three experimental practical diets were formulated with low levels of fish meal (10% FM) and fish oil (7% FO), containing 3 levels of Se. A diet without added Se was used as the control, whereas hydroxy-selenomethionine (OH-SeMet, Adisseo, Selisseo^®^ 2% Se) was supplemented in 2 diets. Thus, the analyzed Se content of the diets was 0.29, 0.52 and 0.79 mg kg^−1^, respectively. The diets were prepared by mixing all the powder and water-soluble components, then the oils and fat-soluble vitamins and, finally, water. The diets were pelleted (California Pellet Mill (CPM, 2HP mod 8.3 USA) and dried in an oven at 40 °C for 24 h and then stored in the fridge until used. All diets were prepared to be isonitrogenous and isoenergetic, the proximate and fatty acid composition were analyzed at GIA laboratories.

### 2.3. Selenium Analysis

Feed samples were first crushed, and an aliquot of 250 mg was digested on a regulated heating block (85 °C for 3 h) with a mixture of nitric acid 69% and hydrogen peroxide and then diluted with ultrapure water. Then, their analysis was performed in the same conditions used for water samples as follows.

The water samples were analyzed by direct injection in the Agilent AT 7500 ICP MS (inductively coupled plasma mass spectrometry) measuring 77Se, 78Se and 82Se isotopes. The quantification was performed by standard addition to avoid any matrix effects. A specific reaction mode with Hydrogen in the CCT (collision cell technology) was activated to stop the polyatomic interferences (with Ar, Cl, Br).

### 2.4. Growth Performance

Once every two weeks and at the end of the feeding trial (13 weeks), individual fish were anesthetized with clove oil (2 mL/100 L) to determine weight, standard length, and total length. Growth performance including the weight gain (WG), feed intake (FI), feed efficiency (FE), specific growth rate (SGR), feed conversion ratio (FCR), protein efficiency ratio (PER), and condition factor were calculated. The equations used are shown below:WG = (final body weight − initial body weight)/initial body weight × 100
FE = (final body weight − initial body weight)/feed intake
SGR = [In (FBW) − In (IBW)]/feeding days × 100
FCR = feed intake (g)/weight gain (g)
CF = Final body weight/standard length^3^ × 100

### 2.5. Biochemical Composition

#### 2.5.1. Proximate Composition

Experimental diets, whole-body and liver samples were frozen at −80 °C until analysis for biochemical composition analysis [29].

#### 2.5.2. Fatty Acid Profiles

The extraction of total lipids from diets and fish whole body and liver was performed by the method of Folch et al. [30] using a mixture of chloroform: methanol (2:1) (*v:v*) containing 0.01% BHT. Lipids were then transmethylated to obtain fatty acid methyl esters (FAMES) described by Christie [31] using nonadecanoic acid (10% of total lipid) as the internal standard. Fatty acid methyl esters were separated by gas liquid chromatography, a flame ionization detector was used to quantification, following the conditions described in Izquierdo et al. [32] and identified by comparison to previously characterized standards.

#### 2.5.3. Fatty Acid Retention Efficiency

The retention efficiency of the most relevant fatty acids was calculated as follows:Relative fatty acid retention (%FA intake) = a − b/c × 100

a: (weight (g) × body lipid (%) × FA (%) in whole body)/100b: (Initial weight (g) × initial body lipid (%) × FA (%) in initial whole body)/100c: Feed intake (g) × (dietary lipid (%) × FA (%) in diet/100) × 100

### 2.6. Crowding Stress Challenge

The crowding test was performed at week 7 of the feeding trial. Initially, five fish were carefully captured from each tank to collect the blood samples in less than 4 min to determine the basal plasma cortisol prior to the crowding challenge. After collecting all the basal plasma cortisol from each tank, another five fish per tank were confined for 2 h in a small cage located in the same tank to determine their response to the crowding challenge. After 2 h of crowding, cages were carefully collected, and blood samples were carefully taken from the caudal sinus by a 1 mL plastic syringe. All the blood samples were immediately transferred to eppendorf with heparin as anticoagulant. The plasma was obtained by centrifugation at 3000 rpm for 10 min and stored at −80 °C prior to cortisol analysis. Cortisol concentration in the perfused fluid was determined by radioimmunoassay (RIA) from Animal Lab (Las Palmas de Gran Canaria, Las Palmas, Spain).

### 2.7. Anti-Viral Response Challenge

At the end of the feeding period, 15 fish per diet were anesthetized in 0.3 mL l-1 2-phenoxiethanol (Panreac, Barcelona, Spain) and intra-peritoneally injected with 100 µL (500 µg) of the double-stranded synthetic RNA, polyinosinic:polycytidylc acid (poly I:C) (Sigma-Aldrich, St. Louis, MI, USA) an immunostimulant extensively used to induce antiviral activities in fish. At 12, 24, 48 and 72 h post-inoculation, fish were euthanized; the individual livers and head kidney samples were immediately preserved in RNAlater (Sigma-Aldrich) for gene expression studies. Total RNA was extracted from seabream liver using TRI Reagent (Sigma-Aldrich) and E.Z.N.A^®^ Total RNA Kit I (OMEGA bio-teck, Norcross, GA, USA). RNA was quantified with NanoDrop 1000 spectrophotometer (Thermo Scientific). Samples were adjusted to the same concentration of 0.5 ng/mL. RNA was reverse transcribed in a 25 µL reaction volume containing 2 µg, using an iScript Reverse Transcription Reagent kit (BioRad, Hercules, CA, USA), until cDNA was obtained in a thermocycler (T100 Therma Cycler, Bio-Rad) following the protocol described by Grasso et al. [33]). The samples were then diluted (1:10) in miliQ water and stored at −20 °C. A selected primer of immune genes was analyzed in IQ5 Multicolor Real-Time PCR detection system (BioRad) using β-actin for housekeeping and SYBR Green Supermix (Biorad). Primers for the gene codifying for the interferon-induced GTP-binding protein Mx (*Mx*), as well as other immune-response-related genes which could be induced by poly I:C, were obtained from Grasso et al. [33] and the qPCR protocol provided by these authors was used to amplify the primers (Table 3).

### 2.8. Statistical Analysis

All results were presented in mean and standard deviation (SD) and were tested for normality and homogeneity of variances using Levene’s test by SPSS 21.0 software (IBM Corp., Chicago, IL, USA). One-way analysis (ANOVA) was used to determine the difference between the diets, multiple means comparisons (Duncan test) were applied when significant differences showed *p* < 0.05. For the plasma cortisol data, a two-way ANOVA was applied to determine the effect of Se level, timing, and the interaction between them. Two regression models (logarithmic and linear regressions analysis) were used to determine the effect of dietary Se on different parameters. Gene expression data were transformed to relative expression against the control non-supplemented diet [34]. For the gene expression analyses, values higher than 1 in a parameter express an increase (up-regulation), while values lower than 1 express a decrease (down-regulation) in a parameter.

## 3. Results

### 3.1. Selenium Analysis

The content of Se in the basal diet (0.29 mg kg^−1^) was proportionally raised by the supplementation with OH-SeMet (0.52 and 0.79 mg kg^−1^) (Table 1). However, the analysis of Se levels in the water showed no significant differences among tanks with fish fed different Se levels along the feeding trial, with average values of 0.95 µg L^−1^ at the beginning of the trial and 1.69 µg L^−1^ after fish feeding (Table 4).

Elevation of dietary Se levels significantly raised Se contents in liver and followed a highly correlated logarithmic regression (y = 0.5357ln(x) + 1.4218, R^2^ = 0.94). A similar trend was found in the Se content in muscle (y = 0.2339ln(x) + 0.4921, R^2^ = 0.72), although no significant differences were found by one-way ANOVA or regression studies.

### 3.2. Growth Performance

Survival was very high along the whole trial (Table 5). All the experimental diets were well accepted by the fish and there were no significant (*p* > 0.05) differences in feed intake among fish fed the different diets. After 13 weeks of the feeding trial, there were no differences in total length, body weight, weight gain, specific growth rate, feed efficiency or condition factor among fish fed the different Se levels (Table 5).

### 3.3. Biochemical Analysis

#### 3.3.1. Proximate Composition

The elevation of dietary Se to 0.52 and 0.79 mg kg^−1^ significantly (*p* < 0.05) increased whole-body lipid (Table 6), following a logarithmic regression with Se contents in diet (y = 1.38ln(x) + 11.19, R^2^ = 0.99) (Figure 1). On the contrary, moisture content in fish fed these diets supplemented with OH-SeMet was significantly lower than those fed the control diet (0.29 mg kg^−1^) (Table 6). Despite whole-body protein contents not being significantly different among fish fed the different Se contents, protein contents tended to increase with the dietary Se content, following a highly correlated logarithmic regression (y = 0.76ln(x) + 17.65; R^2^ = 0. 99) (Table 6) (Figure 1). Whole-body ash content was not affected by dietary Se supplementation.

#### 3.3.2. Fatty Acid Profiles

Dietary essential fatty acid profiles were very similar among the diets with different Se supplementation, with an average of 29.4% saturated fatty acids (SFA), 42% monounsaturated fatty acids (MUFA), 28.4% polyunsaturated fatty acids (PUFA), 13.7% n-3 fatty acids, 14% n-6 fatty acids and 34.7% n-9 fatty acids (Table 2). Fatty acid ratios or the contents in specific fatty acids were neither affected by dietary Se levels.

Whole-body fatty acid profiles were characterized by the abundance in saturated fatty acids, such as stearic acid (18:0), palmitic acid (16:0), oleic acid (OA, 18:1n-9), linoleic acid (LA, 18:2n-6), linolenic acid (18:3n-3), EPA and DHA, regardless the diet fed (Table 7). However, diet supplementation with OH-SeMet significantly (*p* < 0.05) increased PUFA and n-3 PUFA, by 20% and 40%, respectively, in comparison to the fatty acid profiles of whole-body lipid in fish fed the control diet. Moreover, a significant linear regression was found between n-3 PUFA and whole-body lipid contents (y = 3.57x − 21.98, R^2^ = 0.92). This increase in n-3 PUFA was mostly due to the significant (*p* < 0.05) elevation of 20:2n-6, 20:4n-6 (ARA), 20:4n-3, 20:5n-3 (EPA), 22:5n-3 and 22:6n-3 (DHA) in fish fed OH-SeMet. In fish fed OH-SeMet, n-3 fatty acids were increased to a higher extent than n-6 fatty acids and, subsequently, n-3/n-6 and EPA/ARA ratios (Table 7). Interestingly, 18:4n-3, a desaturation product from 18:3n-3, was also significantly increased by the dietary supplementation with OH-SeMet. On the contrary, SFA, MUFA and n-9 fatty acids were significantly (*p* < 0.05) reduced by 12%, 3% and 5% in whole-body lipids of fish fed OH-SeMet supplementation, mainly due to the reduction in 16:0 and 18:1n-9 (Table 7). Moreover, a significant negative logarithmic regression was found between MUFA and whole-body lipid (y = −15.23ln(x) + 78.291, R^2^ = 0.99).

Liver fatty acid profiles followed a similar pattern to the whole-body lipid profiles and, thus, the increase in dietary Se was related to the content of the essential fatty acids (EFA: ARA + EPA + DHA) as well as in the n-3/n-6 and EPA/ARA ratios (Figure 2).

#### 3.3.3. Fatty Acid Retention Efficiency

Fish fed diets supplemented with OH-SeMet showed a 77% higher retention of DHA, 77% of EPA and 52% of n-3 PUFA (*p* < 0.05, Table 8). To a lesser extent than for n-3 fatty acids, there was also a higher retention of ARA (40%) in OH-SeMet fed fish than in control ones. Thus, the EPA/ARA retention was 23% higher in the OH-SeMet fed fish and the n-3/n-6 retention was 36% higher than in control fish. Moreover, the 18:4n-3/18:3n-3 retention was increased by 118% in fish fed OH-SeMet. Indeed, all these retention values followed a highly correlated logarithmic regression with the dietary Se contents (EPA: R^2^ = 0.94; DHA: R^2^ = 0.98; EPA/DHA: R^2^ = 0.8296; n-3/n-6: R^2^ = 0.98; n-3PUFA: R^2^ = 0.97; 18:4n-3/18:3n-3: R^2^ = 0.95) (Table 8).

### 3.4. Crowding Stress Challenge

There were no significant differences in plasma cortisol levels during the crowding stress challenge at 0h (*p* = 0.588) and 2 h (*p* = 0.093) by one-way ANOVA (Table 9). However, after 2 h of crowding confinement, plasma cortisol levels increased compared to the 0 h (two-way ANOVA, *p* = 0.000, Table 9, time effect). Moreover, cortisol levels after 2 h of crowding were significantly increased by the elevation in dietary Se levels compared to 0 h (two-way ANOVA, *p* = 0.013, Table 9, Se level effect). A significant interaction was found between time and Se levels (two-way ANOVA, *p* = 0.017, Table 9, Time x Se level effect). Moreover, plasma cortisol levels after 2 h crowding followed a highly correlated exponential regression with Se contents in diet or liver (relation between cortisol and Se content in liver: y = 212.69 ln(x) + 103.21, R^2^ = 0.77; relation between cortisol and Se content in diet: y = 107.86 ln(x) + 187.05, R^2^ = 0.70).

### 3.5. Anti-Viral Response Challenge

In relation to the initial values at 0 h, inoculation of poly I:C significantly (*p* < 0.01) up-regulated Mx expression after 12 h, regardless of the diet fed (12 h, Figure 3). At 24 h, Mx relative expression values were reduced, particularly in fish fed the control diet, and at 48 h the initial values were recovered (24 h and 48 h, Figure 3).

Comparison of gene expression in fish fed the different diets showed that at 12 h after inoculation there were no significant differences in the expression of Mx in liver, regardless of the diet fed (24 h, Figure 4a). However, at 24 h post-inoculation, the stimulation of Mx expression was significantly higher in fish fed the OH-SeMet-supplemented diets than in those fed the control diet (*p* < 0.05, 24 h, Figure 4a). Moreover, a significant positive linear correlation was found between the dietary Se levels and Mx expression at 24 h (*p* < 0.05, y = 3.252x + 0.1048, R^2^ = 0.96). At 48 h post-inoculation, the same trend was observed, whereas at 72 h the Mx expression values were similar among fish fed the different diets. No significant differences in Mx expression at 48 h or 72 h were found among fish fed the different diets (*p* < 0.05, 24 h, Figure 4a). Despite the fact that, at 12 and 24 h after poly I:C induction, there were no significant differences among fish fed the different diets in the expression of TNF-α and COX, there was a marked up-regulation of both genes at 48 h (Figure 4b, 4c) in fish fed OH-SeMet. However, at 72 h, both TNF-α and COX were up-regulated only in fish fed the highest supplementation of OH-SeMet in comparison to those fed the control diet (*p* < 0.05, 48 h, Figure 4b,c). A similar pattern was observed for IL-Ir2, although no significant differences were detected (Figure 4d). Casp-3 expression was slightly up-regulated by OH-SeMet supplementation at 12 h and 48 h post-poly I:C induction (Figure 4e). However, at 72 h a significant down-regulation of Casp-3 was found in fish fed the Se supplemented diets, particularly in those fed intermediate Se levels (0.52 mg kg^−1^) in comparison to those fed the control diet (*p* < 0.05, 72 h, Figure 4e). For IL-1β, a significant up-regulation of IL-1β was found in fish fed OH-SeMet (*p* < 0.05, 48 h, Figure 4f). There was no clear effect of dietary Se levels in IL-6 or IL-10, except at 72 h when IL-6 seemed to be down-regulated in fish fed OH-SeMet (Figure 4g) and IL-10 was down-regulated in fish fed intermediate Se levels in comparison to those fed the highest supplementation of OH-SeMet (*p* < 0.05, 72 h, Figure 4h).

## 4. Discussion

### 4.1. Effect of Dietary Se in Growth Performance

Dietary supplementation with organic selenium may promote growth in marine fish [3,5,6,8,9]. In the present study, growth performance was not affected by the elevation in dietary Se from 0.29 to 0.79 mg kg^−1^ by supplementation with OH-SeMet, in agreement with previous studies in the same species [7,19,20]. Thus, the growth of gilthead seabream juveniles was not affected by the increase in dietary Se levels from 0.45 to 0.68 mg kg^−1^ as NaSe supplementation [7], from 0.8 to 1.4 mg kg^−1^ as OH-SeMet [20] or from 0.55 to 1.2 mg kg^−1^ as SeMet [19]. In contrast, the weight gain of grouper [3] and the SGR of cobia [5] were reduced when fish were fed with a diet not supplemented with Se, containing 0.21 mg Se kg^−1^ diet, a much lower basal content in Se than studies in seabream. It is noteworthy that a Se content as low as <0.20 mg kg^−1^ was determined in wild gilthead seabream [38], which could also suggest a high tolerance in gilthead seabream to maintain the growth under low Se conditions. Nevertheless, a higher increase in dietary Se supplemented as NaSe to 1 mg kg^−1^ or 1.3 mg kg^−1^, respectively, improves [7] or reduces [20] growth parameters in seabream juveniles. The optimum dietary Se level for seabream is around 0.9 mg kg^−1^; levels higher than 1.3 mg kg^−1^ may be toxic [7,20]. Therefore, the dietary Se range used in the present study was neither severely deficient nor overtly toxic for gilthead seabream.

### 4.2. Effect of Dietary Se in Whole-Body Composition and Se Content in Tissues

The liver is one of the major organs for regulating Se in fish [39]. In the present study, Se contents in liver and muscle were in the range of those early described for this species [20], denoting the good absorption and deposition of Se in the whole body. The increase in dietary Se levels from 0.29 to 0.79 mg kg^−1^ significantly increased Se contents in liver, in agreement with the increase in hepatic Se contents found in previous studies supplementing with either OH-SeMet or NaSe in the same species [7,20] and other species [40]. However, in the present study, Se content in the muscles was lower than in the liver, in agreement with previous studies in gilthead seabream [20], black sea bream (*Acanthopagrus schlegelii*) [40] and Atlantic salmon (*Salmo salar*) [41]. Increasing dietary OH-SeMet from 0.8 to 1.4 mg kg^−1^ significantly increased the Se content in muscle [20], while, in the present study, fed fish by dietary OH-SeMet rising from 0.2to 0.79 mg kg^−1^ were less affected by dietary Se levels. The increase in dietary Se contents, together with a Se increase in body tissues, led to an elevation in whole-body lipid contents, in line with the increase in hepatic lipids in gilthead seabream fed dietary Se to 0.86 mg kg^−1^ as NaSe [7]. Lipid contents are also increased by the elevation of dietary Se by SeMet supplementation from 0.5 to 0.9 mg Se kg^−1^ in rainbow trout [42] and from 0.21 to 1.36 mg Se kg^−1^ in cobia [5]. However, few studies could not find an effect of dietary Se on whole-body or muscle lipid contents in other species [40,43,44].

### 4.3. Effect of Dietary Se in Whole-Body FA’s Composition and Retention

Dietary Se supplementation affects lipid metabolism in fish [45] and mammals [46,47,48,49] and has been associated with an increase in lipogenesis or/and a reduction in lipid catabolism. For instance, intake of high Se in pigs results in elevation of triglycerides, total cholesterol, non-esterified fatty acids, suggesting a net lipogenesis that increases the whole-body lipid content [46]. In the present study, the increase in whole-body lipids was highly related to an increase in n-3 PUFA and a reduction in MUFA and SFA. Therefore, being MUFA and SFA, and particularly 16:0 and 18:1n-9, end-products of lipogenesis in fish, this pathway was not likely responsible for the increased whole-body lipid contents in seabream fed diets supplemented with OH-SeMet. The n-3 PUFA, such as EPA and DHA, are essential fatty acids for marine fish that play very important structural and physiological functions and are synthesized in limited amounts from their 18C precursor (18:3n-3). Delta 6 fatty acid desaturase (Δ6 desaturase) is the first step on n-3 PUFA synthesis from 18:3n-3, where another double bond is introduced to produce 18:4n-3. Subsequently, after several steps of 2-carbon elongation and desaturation (Δ6 desaturase and Δ5 desaturase activities) EPA and DHA are produced [1,50,51]. In the present study, 18:4n-3, product from the first step in n-3 PUFA synthesis, as well as the intermediate products 20:4n-3, EPA, 22:5n-3 and, the final product, DHA were highly correlated with dietary Se. In addition, the ratio of 18:4n-3/18:3n-3 was also significantly higher in the fish fed the Se supplemented diets, further supporting the activation of n-3 PUFA synthesis by Se incorporation into seabream tissues. These results are in agreement with the production of PUFA by their 18C precursors and the up-regulation of related genes in lambs [48] or broiler chickens [49] fed Se supplementations. However, the fatty acids retention in fish altered based on the supplemented nutrient in diet, such as lipid source [52] or mineral [45]. In the present study, fatty acid retention in whole body of seabream showed a retention of DHA higher than 100% for fish fed OH-SeMet diets, demonstrating a net production of this fatty acid and the positive effect of dietary Se levels in PUFA synthesis. These results agree well with the increase in DHA found in sea bass larvae fed a SeMet-supplemented diet [8]. Even though these results point out the role of Se in PUFA synthesis, incorporation of Se into body tissues would also protect these fatty acids from the high risk of peroxidation, contributing to preserve these essential nutrients and its retention in fish tissues. Other minerals also interact with body lipid contents and lipid metabolism in fish or mammals, such as iron [53,54], zinc [55,56], magnesium [57,58] or copper [59,60]. For instance, a marked increase in EPA and DHA retention is found in rainbow trout fed supplemented organic minerals (Zn, Cu, Mn, and Fe) instead of the inorganic forms [45], in relation to the antioxidant role of the first three minerals.

### 4.4. Effect of Dietary Se in Plasma Cortisol

Cortisol is a central stress hormone in fish physiology that is regulated by the hypothalamic–pituitary–interrenal (HPI) axis and is released under stressful conditions such as crowding or handling. Selenium decreases the harmful effects caused by stress, playing a critical role against oxidative cellular injury [1]. Dietary Se supplementation in fish diets can prevent oxidative stress caused from different stressful conditions such as crowding stress [18,20,21], sub-optimal temperature [61], and heavy metal stress [62]. In the present study, basal cortisol levels were low and did not differ among fish fed the different Se levels, denoting the absence of any baseline stress derived from the culture conditions or dietary unbalances [14]. However, cortisol levels were markedly raised after 2 h of crowding. As in other fish species, the post-stress pattern of evolution of plasma cortisol in gilthead seabream is characterized by a maximum cortisol level after 2 h of crowding stress, followed by a decrease after 5 and 24 h [14]. In the present study, at 2 h post-stress, plasma cortisol levels were correlated to Se contents in diets or liver, but were not significantly different, denoting the moderate Se levels used. However, high dietary Se contents over the optimum levels may lead to excessive plasma cortisol levels as observed in rainbow trout (*Salmo gairdneri*) juveniles fed 8.47 mg Se kg^−1^ [63], whereas the optimum level for this species is 0.15 to 0.38 mg kg^−1^ [64]. In the present study, the maximum dietary Se levels (0.79 mg kg^−1^) was in the range of the requirements reported for seabream (0.9 mg kg^−1^). Despite the fact that cortisol release in fish is modulated directly or indirectly by ARA, EPA or DHA [65], no relation was found between the whole-body contents in these fatty acids and the plasma cortisol levels after crowding stress.

### 4.5. Effect of Dietary Se in Gene Expression of Mx Gene and Immune Related Genes

Selenium has been shown to affect both innate and acquired immunity, but its mechanisms of action and its relationship to the immune system are not entirely clear [66]. The innate and acquired immune system is composed of cellular and humoral components that are activated by disease, immunization, or administration of immunostimulants [67]. Selenium may have an immunostimulatory function, but its role is not yet well documented in fish [68,69,70,71]. Feeding pacu (*Piaractus mesopotamicus*) a Se-supplemented diet for 10 days enhances non-specific immune indicators, suggesting that antioxidant status may modulate the immune system [66]. Mx protein is a type I IFN-induced protein that helps inhibit viral RNA translation. As in other studies carried out in vivo for the stimulation of Mx using Poly I: C [67], the expression of this gene showed its highest values during the first days, later decreasing to baseline values. Specifically, in this study in seabream, the highest peak of expression was obtained at 12 h post-stimulation with Poly I: C, regardless of the diet used, recovering the basal expression levels at 48 h, in agreement with previous studies in this species [26]. On the contrary, in Atlantic salmon, the up-regulation of *Mx* gene by induction with Poly I: C remains for up to 9 days [67]. At 24 h post-stimulation, dietary OH-SeMet supplementation significantly raised the expression of the antiviral response gene *Mx*, further extending the protection against viral infections. To our knowledge, this is the first study in gilthead seabream that shows the effect of dietary Se levels on the expression of a Mx protein gene, one of the main components of the antiviral response in vertebrates [72]. The results are in agreement with the up-regulation of *IFN-α* by supplementation of SeMet up to 1 mg kg^−1^, precisely 24 h after the injection of poly I:C in rainbow trout [73], since Mx protein is a IFN-induced GTP-binding protein. Moreover, in trout SeMet supplementation also up-regulated other genes of anti-viral proteins 24 h after poly I:C injection [73] in agreement with the up-regulation of Mx in the present study. Both *TNF-α* and *COX* were also up-regulated by OH-SeMet supplementation, as it occurs in rainbow trout fed SeMet supplemented diets [73]. Other members of the TNF and TNF receptor superfamily and other inflammatory mediators were also up-regulated by SeMet supplementation in trout. Similarly, at 24 h, there was an up-regulation of the pro-inflammatory cytokine *IL-1β*, involved in immune genes regulation. On the contrary, the expressions of *IL-6* and *IL-10*, associated with the anti-inflammatory response, were less affected by dietary Se levels. Overall, these results suggest that Se supplementation causes a stronger pro-inflammatory response after poly I:C induction as an antiviral response. Despite the fact that an extreme pro-inflammatory response could produce unrestrained tissue damage, the high expression levels were recovered at 72 h, particularly in fish fed intermediate Se levels. The expression of the signal transduction mediator *Casp-3* was slightly up-regulated by OH-SeMet supplementation until 48 h post-poly I:C induction, in agreement with the slight enhancement of transcripts for this gene in rainbow trout fed SeMet diets [73]. The immune-regulatory role of Se may be related to its important antioxidant role or their presence in some selenoproteins, such as SeIP, which are important in immune response and, particularly, inflammation [74].

Nevertheless, such an immunostimulatory effect of dietary Se levels could be also indirectly mediated by the proportionally increased the EFAs (ARA, EPA and DHA) contents and the EPA/ARA ratio in liver. Moreover, these fatty acids increased in whole body of fish fed Se supplementation following a significant relation with *Mx* expression at 24 h (ARA: y = 0.0021x + 0.309, R^2^ = 0.96; EPA: y = 0.0247x + 1.4822, R^2^ = 0.95; DHA: y = 0.0577x + 2.375, R^2^ = 0.93). Both the type and level of dietary PUFA regulate the immune system [11] and the mechanisms involved in the modulation of the immune response by PUFA are well studied in mammals and, to a lesser extent, in fish [11,15]. Thus, changes in dietary PUFA alter fatty acid profiles of immune cells affecting membrane fluidity, intercellular interaction, receptors expression, or signal transductions. Additionally, these PUFAs produce certain eicosanoids and docosanoids that can either stimulate or inhibit immune cells in a dose-dependent manner. In fact, both classical (prostaglandins, PG; leukotrienes, LT; thromboxane, TXa) and non-classical (resolvins, maresins and protectins) eicosanoids and docosanoids [75] derived from ARA, EPA and DHA have important immunological functions [75]. These highly bioactive compounds act as pro-resolving mediators and recover homeostasis by resolving inflammatory response [76]. Particularly ARA and EPA are excellent substrates for COX enzymes, each leading to eicosanoids with different activity capacity. In gilthead seabream a recent study also demonstrates the importance to balance the dietary contents in ARA, EPA and DHA to better modulate the fish innate immune response to against potential immunological insults in gilthead seabream juveniles [77]. More specifically, DHA regulates the expression of IFN-induced genes expression in mammals [78] and could contribute for the up-regulation of *Mx* gene in the present study. Hence, the OH-MetSe-supplemented diets which led to a higher retention of EPA, DHA, and ARA in body tissues may help protect against viral infections through IFN-induced antiviral and resolve, helping to resolving inflammation.

## 5. Conclusions

Elevation in dietary Se from 0.29 to 0.79 mg kg^−1^ by supplementation with OH-SeMet, proportionally raised Se contents in body tissues and increased lipid contents in the whole body, promoting retention and synthesis of n-3 PUFA and a reduction in MUFA and SFA. Specifically, a net production of DHA was observed in those fish fed with diets supplemented with OH-SeMet. Despite the fact that the elevation of dietary Se to 0.79 mg kg^−1^ did not affect basal cortisol levels, denoting the absence of chronic stress due to nutritional unbalances, it markedly increased 2 h post-stress plasma cortisol levels, denoting the moderate Se levels used. Finally, at 24 h post-stimulation, dietary OH-SeMet supplementation significantly raised the expression of the antiviral response gene *Mx*, as well as other immune system related genes, showing, for the first time in gilthead seabream, the importance of dietary Se levels in antiviral response.

## Figures and Tables

**Figure 1 animals-11-02877-f001:**
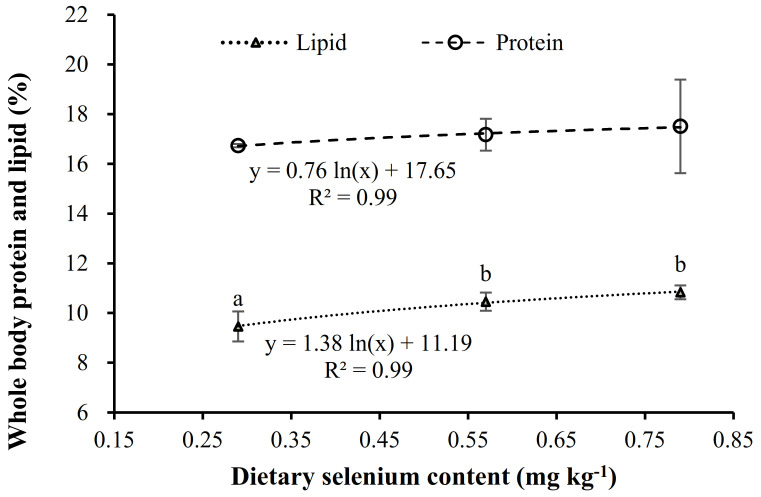
Relation between dietary Se levels and protein or lipid contents (% fresh weight) in whole body of gilthead seabream after 13 weeks of feeding diets with different supplementation of OH-SeMet.

**Figure 2 animals-11-02877-f002:**
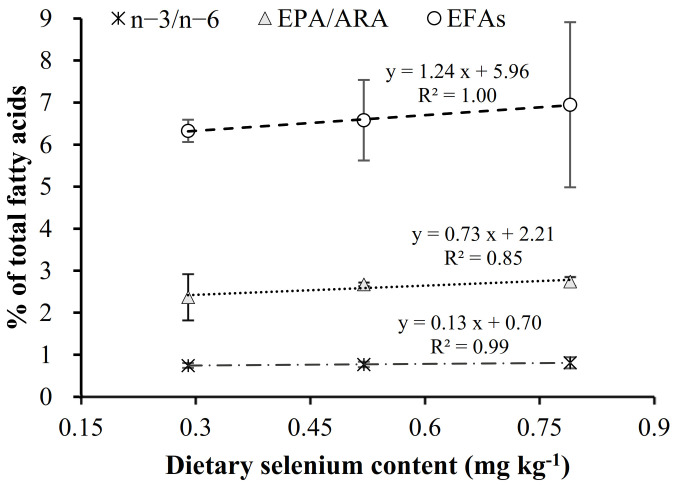
Relation between dietary Se levels and n-3/n-6, EPA/ARA and EFAs (essential fatty acids: DHA + EPA + ARA) in liver of gilthead seabream after 13 weeks of feeding diets with a different supplement of OH-SeMet.

**Figure 3 animals-11-02877-f003:**
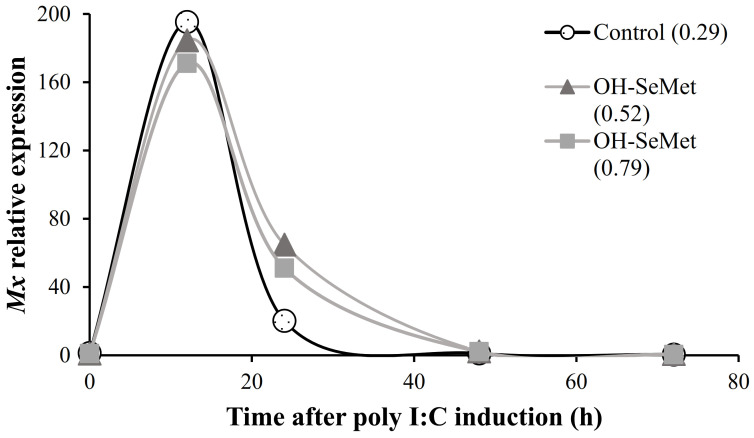
Relative expression of *Mx* (fold changes in relation to the values at 0 h for each diet) in liver of gilthead seabream fed diets with different supplementation of OH-SeMet along the anti-viral response challenge.

**Figure 4 animals-11-02877-f004:**
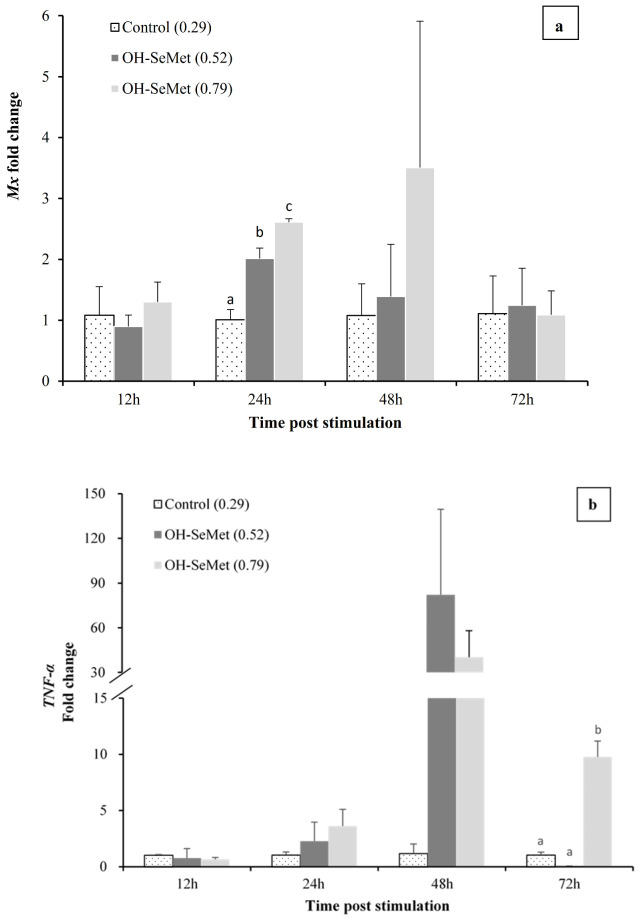
Relative expression (fold changes in relation to the values of the control diet) of selected immune response genes in liver ((**a**): *Mx*) and head kidney ((**b**): *TNF-α*; (**c**): *COX*; (**d**): *IL-Ir2*; (**e**): *Casp-3*; (**f**): *IL-1β*; (**g**): *IL-6*; (**h**): *IL-10*) of gilthead seabream fed diets with different supplementation of OH-SeMet along the anti-viral response challenge.

**Table 1 animals-11-02877-t001:** Ingredients and proximate composition of the experimental diets containing different Se levels.

Ingredient (%)	Diets		
	Control (0.29)	OH-SeMet (0.52)	OH-SeMet (0.79)
Fish meal ^a^	10.00	10.00	10.00
Blood meal ^b^	6.00	6.00	6.00
Rapeseed meal ^a^	12.70	12.70	12.70
Corn gluten meal ^a^	18.40	18.40	18.40
Soy protein concentrate	18.30	18.30	18.30
Wheat meal	4.72	4.72	4.72
Wheat gluten	6.00	6.00	6.00
Fish oil	7.00	7.00	7.00
Vegetable oil mix ^c^	10.00	10.00	10.00
Vitamin mix	2.00	2.00	2.00
Mineral mix	2.00	2.00	2.00
OH-SeMet ^d^	-	0.00125	0.0025
Ca(H_2_PO_4_)_2_	1.62	1.62	1.62
L-Lysine	0.46	0.46	0.46
DL-Methionine	0.06	0.06	0.06
L-Histidine	0.14	0.14	0.14
Cholesterol	0.11	0.11	0.11
CMC	0.50	0.50	0.50
Proximate composition (%)			
Ash % (DM)	6.98	6.96	7.05
Total lipids% (DM)	21.21	21.85	22.48
Crude protein% (DM)	48.20	48.05	48.23
Analysed Se content			
(mg kg^−1^)	0.29	0.52	0.79

^a^ Dibaq (Spain); ^b^ Porcine origin, (Dibaq) Spain; ^c^ Vegetable oil mix: colza (4%), linseed (2%) and palm (4%); ^d^ Selenium sources: OH-SeMet, hydroxy-selenomethionine (Adisseo, Selisseo^®^ 2% Se).

**Table 2 animals-11-02877-t002:** Fatty acid composition of the experimental diets containing different Se levels (% total identified fatty acids).

Fatty Acids	Diets		
	Control (0.29)	OH-SeMet (0.52)	OH-SeMet (0.79)
11:0	0.16	0.13	0.13
12:0	0.46	0.45	0.46
13:0	2.66	2.82	2.92
14:0	2.17	2.21	2.21
14:1n-5	0.07	0.07	0.07
15:0	0.24	0.24	0.24
16:0	18.76	18.52	19.10
16:1n-7	2.28	2.34	2.32
16:1n-5	0.10	0.11	0.11
16:2n-4	0.17	0.18	0.17
17:0	0.14	0.14	0.13
16:3n-4	0.16	0.15	0.16
16:3n-3	0.07	0.07	0.07
16:4n-3	0.17	0.18	0.16
18:0	4.14	4.13	4.26
18:1n-9	34.46	33.87	34.47
18:1n-7	2.32	2.33	2.32
18:1n-5	0.07	0.07	0.07
18:2n-6	13.64	13.43	13.25
18:2n-4	0.06	0.06	0.06
18:3n-6	0.08	0.09	0.08
18:3n-4	0.05	0.06	0.06
18:3n-3	6.99	6.87	6.52
18:4n-3	0.37	0.41	0.36
18:4n-1	0.04	0.04	0.04
20:0	0.41	0.40	0.42
20:1n-9	0.15	0.16	0.16
20:1n-7	1.51	1.52	1.54
20:1n-5	0.11	0.11	0.12
20:2n-9	0.09	0.18	0.20
20:3n-9	0.05	0.05	0.05
20:4n-6	0.32	0.34	0.31
20:3n-3	0.07	0.07	0.07
20:4n-3	0.20	0.21	0.20
20:5n-3	2.24	2.45	2.17
22:1n-11	1.01	1.03	1.05
22:1n-9	0.23	0.23	0.24
22:4n-6	0.05	0.05	0.05
22:5n-6	0.02	0.02	0.02
22:5n-3	0.44	0.49	0.43
22:6n-3	3.13	3.57	3.08
∑SFA ^§^	29.17	29.08	29.90
∑MUFA ^§^	42.10	41.62	42.25
∑PUFA ^§^	28.50	29.07	27.61
∑n-3	13.67	14.32	13.07
∑n-6	14.17	14.00	13.79
∑n-9	34.76	34.28	34.90
n-3/n-6	0.96	1.02	0.95
DHA/EPA ^§^	1.40	1.46	1.42
EPA/ARA ^§^	7.03	7.19	6.89

^§^ SFA: saturated fatty acid; MUFA: monounsaturated fatty acid; PUFA: polyunsaturated fatty acid; DHA: docosahexaenoic acid; EPA: eicosapentaenoic acid; ARA: arachidonic acid.

**Table 3 animals-11-02877-t003:** Primers used to assess the immune system in qPCR.

Gene	Primers Sequence	Pmol/µL	Reference
*β-actin*	Forward 5′-TCT GTC TGG ATC GGA GGC TC-3′	10	Grasso et al. [33]
	Reverse 5′-AAG CAT TTG CGG TGG ACG-3′		
*Mx*	Forward 5′-GAC AGG GAG CGG CAT TGT TAC-3′	10	Grasso et al. [33]
	Reverse 5′-TCG TCC AGC TCT TCC TCG TG-3′		
*TNF-a*	Forward 5′-TCG TTC AGA GTC TCC TGC AG-3′	10	Grasso et al. [33]
	Reverse 5′-CAT GGA CTC TGA GTA GCG CGA-3′		
*COX*	Forward 5′-GAG TAC TGG AAG CCG AGC AC-3′	10	Grasso et al. [33]
	Reverse 5′-GAT ATC ACT GCC GCC TGA CT-3′		
*IL-Ir2*	Forward 5′-AAG GAC TCC AGC TCC ACT GA-3′	10	Grasso et al. [33]
	Reverse 5′-ACG CCT TCT ACA TGG ACC AC-3′		
*Casp-3*	Forward 5′-CTGATCTGGATGGAGGCATT-3′	10	Morcillo et al. [35]
	Reverse 5′-AGTAGTAGCCTGGGGCTGTG-3′		
*IL-1β*	Forward 5′-AGC GAC ATG GCA CGA TTT-3′	10	Grasso et al. [33]
	Reverse 5′-GCA CTC TCC TGG CAC ATA TCC-3′		
*IL-6*	Forward 5′-GCT CTG CTG GGT GTG CTC C-3′	10	Castellana et al. [36]
	Reverse 5′-GTC TCC CAC TCC TCA CCT TG-3′		
*IL-10*	Forward 5′-TGGAGGGCTTTCCTGTCAGA-3′	10	Pellizzari et al. [37]
	Reverse 5′-TGCTTCGTAGAAGTCTCGGATGT-3′		

**Table 4 animals-11-02877-t004:** Selenium content in tank water (µg L^−1^) and body tissues (mg kg^−1^) of gilthead seabream after 13 weeks of feeding diets with different supplementation of OH-SeMet.

	Diets		
Se Content in Water (µg L^−1^)	Control (0.29)	OH-SeMet (0.52)	OH-SeMet (0.79)
Before feeding (0 day)	1.14 ± 0.06	0.78 ± 0.15	0.92 ± 0.54
After feeding (4 day)	1.72 ± 0.09	1.69 ± 0.28	1.67 ± 0.29
Se content in body tissues (mg kg^−1^)			
Liver	0.79 ± 0.02 ^a^	1.04 ± 0.08 ^a,b^	1.35 ± 0.21 ^b^
Muscle	0.18 ± 0.01	0.45 ± 0.19	0.38 ± 0.04

Different superscripts denote significant (*p* < 0.05) differences among fish from different dietary treatments.

**Table 5 animals-11-02877-t005:** Survival, growth performance and feed intake in gilthead seabream after 13 weeks of feeding diets with different supplementation of OH-SeMet.

Diets	Control (0.29)	OH-SeMet (0.52)	OH-SeMet (0.79)
Survival rate (%)	97.30 ± 3.82	98.65 ± 1.91	97.30 ± 3.82
Total length (cm)	15.22 ± 0.27	15.22 ± 0.18	15.28 ± 0.14
Body weight (g)	51.54 ± 4.19	49.68 ± 1.19	50.89 ± 2.57
WG ^a^ (%)	135.77 ±18.20	126.95 ± 3.44	135.50 ± 9.17
SGR (%)	0.93 ± 0.08	0.89 ± 0.02	0.93 ± 0.04
FI (g)	52.00 ± 4.68	51.01 ± 5.08	55.21 ± 2.39
FE	0.57 ± 0.03	0.55 ± 0.03	0.53 ± 0.02
CF	1.91 ± 0.07	1.86 ± 0.01	1.43 ± 0.03

^a^ WG: weight gain; FI: feed intake; FE: feed efficiency; SGR: specific growth rate; CF: Condition factor (CF). Different superscripts denote significant (*p* < 0.05) differences among fish from different dietary treatments.

**Table 6 animals-11-02877-t006:** Whole-body proximate composition (% fresh weight) of seabream after 13 weeks of feeding diets with different supplementation of OH-SeMet.

	Diets		
(% Fresh Weight)	Control (0.29)	OH-SeMet (0.52)	OH-SeMet (0.79)
Lipid	9.46 ± 0.60 ^a^	10.45 ± 0.37 ^b^	10.83 ± 0.65 ^b^
Moisture	68.61 ± 0.85 ^b^	65.75 ± 2.64 ^a^	64.86 ± 2.77 ^a^
Protein	16.73 ± 0.07	17.17 ± 0.64	17.51 ± 1.88
Ash	3.80 ± 0.31	3.99 ± 0.43	4.02 ± 0.44

Different superscripts denote significant (*p* < 0.05) differences among fish from different dietary treatments.

**Table 7 animals-11-02877-t007:** Fatty acids composition of total lipids from whole body of gilthead seabream fed diets with different supplementation of OH-SeMet (% total identified fatty acids).

Diets			
Fatty Acids	Control (0.29)	OH-SeMet (0.52)	OH-SeMet (0.79)
11:0	0.07 ± 0.03	0.05 ± 0.01	0.05 ± 0.03
12:0	0.13 ± 0.01 ^b^	0.11 ± 0.00 ^a^	0.13 ± 0.00 ^ab^
13:0	6.92 ± 0.73	4.58 ± 0.99	5.70 ± 1.45
14:0	2.83 ± 0.05	2.79 ± 0.01	2.65 ± 0.12
14:1n-7	0.05 ± 0.00	0.05 ± 0.00	0.05 ± 0.00
14:1n-5	0.08 ± 0.00	0.08 ± 0.00	0.07 ± 0.00
15:0	0.28 ± 0.01 ^b^	0.27 ±0.00 ^a^	0.26 ± 0.01 ^a^
16:0	15.53 ± 0.29 ^b^	14.53. ± 0.12 ^a^	14.47 ± 0.32 ^a^
16:1n-7	4.14 ± 0.06	4.27 ± 0.03	4.09 ± 0.20
16:1n-5	0.11 ± 0.00	0.11 ± 0.00	0.11 ± 0.00
16:2n-4	0.23 ± 0.01 ^a^	0.26 ± 0.00 ^b^	0.25 ± 0.01 ^b^
17:0	0.22 ± 0.08	0.22 ± 0.00	0.22 ± 0.01
16:3n-4	0.25 ± 0.06	0.22 ± 0.01	0.22 ± 0.00
16:3n-3	0.07 ± 0.00	0.07 ± 0.00	0.07 ± 0.00
16:3n-1	0.07 ± 0.00	0.08 ± 0.00	0.07 ± 0.00
16:4n-3	0.13 ± 0.02 ^a^	0.21 ± 0.01 ^b^	0.20 ± 0.01 ^b^
18:0	3.46 ± 0.09 ^b^	3.37 ± 0.03 ^b^	3.24 ± 0.05 ^a^
18:1n-9	32.90 ± 0.62 ^b^	31.29 ± 0.36 ^a^	31.16 ± 0.81 ^a^
18:1n-7	2.84 ± 0.04 ^b^	2.83 ± 0.03 ^b^	2.73 ± 0.02 ^a^
18:1n-5	0.11 ± 0.00	0.12 ± 0.00	0.11 ± 0.01
18:2n-9	0.37 ± 0.04 ^a^	0.40 ± 0.04 ^ab^	0.47 ± 0.03 ^b^
18:2n-6	12.11 ± 0.15	12.25 ± 0.14	12.19 ± 0.35
18:2n-4	0.03 ± 0.01 ^a^	0.02 ± 0.00 ^a^	0.04 ± 0.00 ^b^
18:3n-6	0.12 ± 0.01	0.13 ± 0.00	0.13 ± 0.00
18:3n-4	0.11 ± 0.00 ^a^	0.13 ± 0.01 ^c^	0.12 ± 0.00 ^b^
18:3n-3	4.11 ± 0.16	4.32 ± 0.13	4.48 ± 0.37
18:4n-3	0.49 ± 0.07 ^a^	0.70 ± 0.02 ^b^	0.71 ± 0.01 ^b^
18:4n-1	0.06 ± 0.01 ^a^	0.09 ± 0.00 ^b^	0.09 ± 0.00 ^b^
20:0	0.28 ± 0.01 ^b^	0.26 ± 0.00 ^a^	0.26 ± 0.01 ^a^
20:1n-9	0.29 ± 0.01	0.31 ± 0.00	0.29 ± 0.02
20:1n-7	1.82 ± 0.07	1.80 ± 0.04	1.73 ± 0.08
20:1n-5	0.14 ± 0.00	0.14 ± 0.00	0.14 ± 0.00
20:2n-9	0.23 ± 0.01	0.22 ± 0.03	0.25 ± 0.02
20:2n-6	0.36 ± 0.02 ^a^	0.42 ± 0.00 ^b^	0.40 ± 0.01 ^b^
20:3n-9	0.01 ± 0.00	0.02 ± 0.00	0.02 ± 0.00
20:3n-6	0.15 ± 0.00 ^a^	0.16 ± 0.01 ^b^	0.17 ± 0.01 ^b^
20:4n-6	0.35 ± 0.04 ^a^	0.44 ± 0.01 ^b^	0.43 ± 0.01 ^b^
20:3n-3	0.17 ± 0.01 ^a^	0.19 ± 0.00 ^b^	0.19 ± 0.00 ^b^
20:4n-3	0.35 ± 0.04 ^a^	0.40 ± 0.03 ^b^	0.40 ± 0.01 ^b^
20:5n-3	1.93 ± 0.32 ^a^	2.98 ± 0.11 ^b^	2.90 ± 0.10 ^b^
22:1n-11	1.13 ± 0.06	1.15 ± 0.02	1.09 ± 0.06
22:1n-9	0.43 ± 0.02	0.42 ± 0.00	0.42 ± 0.02
22:4n-6	0.10 ± 0.00 ^a^	0.11 ± 0.00 ^b^	0.11 ± 0.00 ^b^
22:5n-6	0.03 ± 0.00	0.03 ± 0.00	0.03 ± 0.01
22:5n-3	0.88 ± 0.13 ^a^	1.41 ± 0.06 ^b^	1.36 ± 0.03 ^b^
22:6n-3	3.41 ± 0.65 ^a^	5.84 ± 0.26 ^b^	5.73 ± 0.07 ^b^
∑SFA *	29.77 ±1.02 ^b^	26.23 ± 0.88 ^a^	26.92 ± 1.23 ^a^
∑MUFA *	44.07 ± 0.64 ^b^	42.58 ± 0.45 ^a^	42.00 ± 0.60 ^a^
∑PUFA *	26.16 ±1.58 ^a^	31.18 ± 0.55 ^b^	31.09 ± 0.64 ^b^
∑n−3	11.56 ±1.40 ^a^	16.16 ± 0.48 ^b^	16.08 ± 0.33 ^b^
∑n−6	13.22 ± 0.21	13.55 ± 0.13	13.46 ± 0.32
∑n−9	34.23 ± 0.64 ^b^	32.67 ± 0.35 ^a^	32.60 ± 0.80 ^a^
n−3/n−6	0.87 ± 0.09 ^a^	1.19 ± 0.04 ^b^	1.19 ± 0.01 ^b^
DHA/EPA *	1.76 ± 0.05 ^a^	1.96 ± 0.02 ^b^	1.98 ± 0.07 ^b^
EPA/ARA *	5.54 ± 0.40 ^a^	6.82 ± 0.08 ^b^	6.79 ± 0.10 ^b^
18:4n-3/18:3n-3	0.12 ± 0.01 ^a^	0.16 ± 0.01 ^b^	0.16 ± 0.02 ^b^

* SFA: saturated fatty acid; MUFA: monounsaturated fatty acid; PUFA: polyunsaturated fatty acid; DHA: docosahexaenoic acid; EPA: eicosapentaenoic acid; ARA: arachidonic acid. Different superscripts denote significant (*p* < 0.05) differences among fish from different dietary treatments.

**Table 8 animals-11-02877-t008:** Fatty acid retention (% FA intake) in whole-body lipids of gilthead seabream after 13 weeks of feeding diets with different supplementation of OH-SeMet.

	Diets			R^2^	*p* Value
Fatty Acid	Control (0.29)	OH-SeMet (0.52)	OH-SeMet (0.79)		
20:4 n-6	47.04 ± 8.61 ^a^	65.65 ± 1.08 ^b^	68.16 ± 9.79 ^b^	0.94	0.027
20:5 n-3	41.51 ± 8.98 ^a^	70.78 ± 1.01 ^b^	76.10 ± 8.20 ^b^	0.95	0.002
22:6 n-3	57.12 ± 13.65 ^a^	101.19 ± 1.45 ^b^	113.46 ± 15.82 ^b^	0.98	0.003
n-3 LC-PUFA	44.59 ± 6.88 ^a^	67.54 ± 1.68 ^ab^	72.91 ± 11.20 ^b^	0.97	0.009
DHA/EPA	1.37 ± 0.26 ^a^	1.43 ± 0.16 ^ab^	1.49 ± 0.29 ^b^	0.98	0.026
EPA/ARA	0.88 ± 0.19 ^a^	1.08 ± 0.06 ^b^	1.12 ± 0.08 ^b^	0.96	0.009
18:4 n-3/18:3 n-3	1.19 ± 0.04 ^a^	2.60 ± 0.01 ^b^	2.72 ± 0.05 ^b^	0.95	0.013
n-3/n-6	1.07 ± 0.10 ^a^	1.45 ± 0.02 ^b^	1.58 ± 0.04 ^b^	0.98	0.006
18:3 n-6/18:2 n-6	0.95 ± 0.18	1.12 ± 0.07	1.20 ± 0.07	0.99	0.096

Different superscripts denote significant (*p* < 0.05) differences among fish from different dietary treatments.

**Table 9 animals-11-02877-t009:** Plasma cortisol levels (ng mL^−1^) in gilthead seabream fed diets with different supplementation of OH-SeMet at 0 and 2 h after the crowding stress challenge.

	Diets			One-Way ANOVA		Two-Way ANOVA	
Plasma Cortisol (ng mL^−1^)	Control (0.29)	OH-SeMet (0.52)	OH-SeMet (0.79)	*p* Value	Time	Se Level	Time × Se Level
0 h	8.39 ± 1.40	8.57 ± 1.83	10.75± 4.65	0.588	0.000	0.013	0.017
2 h	70.75± 37.89	75.25 ± 1.77	185.67 ± 71.65	0.093			

## Data Availability

The data presented in this study are available from the corresponding author on request.

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
