# Peer review of "Organic Selenium (OH-MetSe) Effect on Whole Body Fatty Acids and Mx Gene Expression against Viral Infection in Gilthead Seabream (Sparus aurata) Juveniles"

_animals, 2021, doi:10.3390/ani11102877_

Round 1

Reviewer 1 Report

The manuscript is very well structured, with consistent data on an important topic in fish nutrition. As for the form, I suggest improving the conclusion so as not to repeat sentences in the results.
My opinion is that, as small changes, the mauscript can be accepted.

Author Response

Thank you very much for your time to review our manuscript.

Reviewer 2 Report

The authors investigated in this study the effects of dietary supplementation of OH-SeMet on growth performance, body fatty acid profile, and immune response in gilthead seabream juveniles.

The manuscript is well structured and organized and the analytical methods used were adequate to obtain the data to accomplish the study objectives, which are of interest and fit well within the scope of the Journal.

Τhe numbering of the lines is missing entirely in the first 15 pages, making the reviewers’ task strenuous.

Some suggestions for minor improvements are provided below:

Abstract

Improve English in the sentence “Moreover, …viral infections’’.

Add ‘growth performance’ to the aim of the study as well.

'0.79 mg kg-1'

Use ‘although’ instead of ‘despite’

‘Se content’ instead of ‘Se contents’. Similarly, for ‘lipid contents’

Finally, at 24h post-stimulation.

‘significantly increased’ instead of ‘significantly raised’.

Introduction

‘no studies have been focused on the Se effects against viral infections’.

How do you follow the EFSA recommendation since you have used 0.52 and 0.79 mg kg-1 Se in the diets when the recommended supplementation by EFSA is 0.5 mg kg-1?

Materials and Methods

Please explain better what do you mean by feeding the fish with approximately 25% of the daily ration.

‘All diets were prepared in isonitrogenous and isolipid’ is redundant.

Please define the dose for clove oil.

In statistical analysis, explain why you have selected p < 0.10- for Se content.

‘A similar trend was found in Se content in muscle’. Please rephrase, there was no similar trend.

In the proximate composition section, please check the regression equation between body lipid and Se.

‘Whole body fatty acid profiles were characterized…’

In the crowding stress challenge, please check again your statistics; it seems strange to have significant differences with such a large deviation among the treatments. Furthermore, why define p = 0.06 as significant?

Discussion

Line 41: could you mention references on other fish species regarding Se content in the muscle?

Line 88: ‘after 2h’

Line 100: please insert reference.

Line 107: ‘its role’.

Lines 135-142: please improve the English of this passage.

Author Response

(The authors gave the same response as above.)

Reviewer 3 Report

The authors investigated the effect of organic selenium (OH-MetSe) effect on whole-body fatty acids and immune response against viral infections in gilthead seabream. They designed three treatments to test their hypothesis regarding the selenium impact on lipid metabolism and stress response. This manuscript (MS) was clearly written and easy to understand. This work can help the sustainability of this species farming as few studies have been done on this topic. However, some issues significantly compromised the quality of this MS.

Abstract

  • Line 15-16, this sentence needs a reference; it is better to delete it and add something more general.
  • Line 19-21, this cause misconception and readers think you did not have control diet. Actually, you supplemented Se with two diets.
  • Line 19, please delete the fish weight and dosage of Se here.
  • Please make sure you defined the abbreviations in the MS for the first time.
  • You measured only one gene, so it is better to mention the complete name of Mx here and elsewhere.
  • Line 26-28, please revise this sentence.
  • Line 36-37, same comment as I mentioned in lines 19-21
  • Change to: “this infection”.
  • Change “d” to days
  • It is better to not start the sentences with abbreviations.
  • Line 26-28, this is not correct; hips of studies have been done to mitigate the stress response which means that reducing the cortisol after stress. If the fish fed a higher dose of Se and had higher cortisol after stress, it is not a mitigation effect and even has caused more stress.
  • Line 27, It is not for the first time; please search the literature more comprehensively. Please delete this claim throughout the MS.
  • Please reorder the keywords alphabetically and capitalize each word.
  • Please write the abstract more numerically about the results. You can do it by adding their numbers in parentheses.
  •  
  • Introduction:
  • Well-developed introduction and included a clear fellow and relevant points.
  • Line 55, Please here and elsewhere, focus on fish and numerous studies have been done in aquatic animals and no need to cite references from humans or other animals.
  • Line 56-63, summarize this part and say Se has an important role in the antioxidant system.
  • Line 62-67, this part is not relative to your study as you did not measure Se deficiency. You actually added Se more than a normal level. Mentioning to Se requirement of this fish species will illustrate this point.
  • Throughout the MS, please first mention the common name plus scientific name, and for the rest of the MS, just report the common name.
  • Line 102-104, Is it relative to this study? I suggest only focus on MX as you measured in this study and delete them.
  • Line 125, please add how much is that.
  • Please update the introduction with recent works as many studies are available from the last two years, which were not included in this section.

Material and methods

  • Well-organized section. Clear fellow and all required details were provided.
  • How much clove oil?
  • Line 195, dietary fatty acids?
  • Statistical analysis, doing polynomial regression is required for this study along with ANOVA to understand changes. It seems that all of them had linear relations. However, please do this analysis and state that somewhere, there was no quadratic relation, and all of them were linear.
  •  

Results

  • Throughout the MS, if there is no significant difference, no need to report P-value.
  • It is a question for me how is possible they have 97% correlation with each other, but there are not significantly different based on ANOVA; please double-check them.
  • Lines 271-275, No need to report this as you did not add any lipid-based ingredients, and not surprisingly, they are the same.
  • Line 276, please summarise this part and mention only important results.
  • I suggest deleting the regression formulas from this MS as they have not added anything new to the concept and made the results, tables and plots too busy. You can only keep the R2.
  • Line 306, what is poly I:C?
  • Table 2 and Table 7, please delete the fatty acids with a value less than 0.4. Even some researchers believe we should not report those less than 1 %.
  • I suggest using the complete name of MX throughout the MS and tables.
  • Table 4, please bold “Se content in body tissues”
  • Line 334 and elsewhere, please mention it in a complete form “gilthead sea bream”.
  • Line 360, Please somewhere is discussion mention how much retention is normal for this species.
  • Please move table 8 to the supplementary file as you already mentioned to this information in Figure 2.
  • Please report Table 9 in a plot.
  •  

Discussion

  • Put the subheading for the discussion section like results. Also, keep a sequence in subheading for investigated factors, in M%M, result, and discussion.
    • Line 394, please keep the consistency of mg/kg in the MS.
    • Line 458-460 is incorrect and shows that Se has not been useful for stress and did not have an alleviation effect.
    • Line 485- 507, Please keep this point in mind that all changes in MX are due to feeding fish with Se and not EPA and DHA. Although these fatty acids change, but it is not completely correct to connect them to the Mx changes. Mx changes and EPA, DHA alterations are all the consequences of Se (cause and effect).

Best regards

Author Response

(The authors gave the same response as above.)

Round 2

Reviewer 3 Report

The authors improved the quality of the MS, and I suggest authors reading one more time to fix few language errors. Then, it would be ready for the final steps for acceptance.

  • Please make sure scientific names are italic; I can see some errors in the MS.

Best regards
